# Utilisation of semiconductor sequencing for detection of actionable fusions in solid tumours

**Marco Loddo** *, **Keeda-Marie Hardisty, Alexander Llewelyn, Tiffany Haddow,**
**Robert Thatcher, Gareth Williams** *

Oncologica UK Ltd, Cambridge, United Kingdom

* marco.loddo@oncologica.com (ML); gareth.williams@oncologica.com (GW)

**Data Availability Statement:** All relevant data are within the manuscript and its Supporting information files.

## Abstract

Oncogenic fusions represent compelling druggable targets in solid tumours highlighted by the recent site agnostic FDA approval of larotrectinib for NTRK rearrangements. However screening for fusions in routinely processed tissue samples is constrained due to degradation of nucleic acid as a result of formalin fixation., To investigate the clinical utility of semiconductor sequencing optimised for detection of actionable fusion transcripts in formalin fixed samples, we have undertaken an analysis of test trending data generated by a clinically validated next generation sequencing platform designed to capture 867 of the most clinically relevant druggable driver-partner oncogenic fusions. Here we show across a real-life cohort of 1112 patients with solid tumours that actionable fusions occur at high frequency (7.4%) with linkage to a wide range of targeted therapy protocols including seven fusion-drug matches with FDA/EMA approval and/or NCCN/ESMO recommendations and 80 clinical trials. The more prevalent actionable fusions identified were independent of tumour type in keeping with signalling via evolutionary conserved RAS/RAF/MEK/ERK, PI3K/AKT/MTOR, PLCy/PKC and JAK/STAT pathways. Taken together our data indicates that semiconductor sequencing for detection of actionable fusions can be integrated into routine diagnostic pathology workflows enabling the identification of personalised treatment options that have potential to improve clinical cancer management across many tumour types.

## Introduction

Oncogenic fusion genes are an important class of driver mutation in solid tumours and haematological malignancies [1]. The rationale for targeting gene fusions was first highlighted by the significant clinical responses to imatinib in patients with BCR-ABL-positive chronic myeloid leukaemia [2]. This therapeutic approach has now been extended to solid tumours, for example the targeting of ALK, ROS1, NTRK and RET fusion genes in non-small cell lung cancer (NSCLC) where the clinical impact has been far reaching with markedly improved survival outcomes [3, 4].

For decades, the major pathway for development and approval of oncology drugs has centred on histopathological parameters, anatomical location and clinical data from all-comers

**Funding:** The authors received no specific funding for this work. Data analysis conducted in this study was limited to secondary use of information previously collected in the course of normal care. No dedicated funding source was allocated for this study. The funder provided support in the form of salaries for all authors, but did not have any additional role in the study design, data collection and analysis, decision to publish, or preparation of the manuscript. The specific roles of these authors are articulated in the 'author contributions' section.

**Competing interests:** The authors have read the journal's policy and the authors of this manuscript have the following competing interests: Authors ML, KH, AL, RT, and GW are currently salaried employees at Oncologica UK Ltd. TH was a salaried employee at the time of data generation for this manuscript. There are no patents, products in development, or marketed products associated with this research to declare. This does not alter our adherence to PLOS ONE policies on sharing data and materials.

clinical trials [5]. However, the ecosystem for drug development has now changed dramatically with the recent landmark FDA approval for larotrectinib targeting neurotrophic receptor tyrosine kinase (NTRK) oncogenic fusions and pembrolizumab for tumours exhibiting microsatellite instability (MSI) and/or mismatch repair (MMR) defects. These drugs have been approved based on efficacy linked to specific molecular aberrations and not on conventional clinico-pathological parameters [6, 7]. This new functional tissue agnostic approach to targeted therapies and immunotherapies is being rapidly accelerated by the expansion of molecular basket clinical trials in which detection of an actionable genetic variant is used as the determinant for entry into a clinical trial [8, 9].

Although many oncogenic fusion genes have now been identified, few are screened as potential therapeutic targets in routine clinical practice. Testing is mostly restricted to the detection of ALK and ROS rearrangements in NSCLC [10]. This relates to the fact that fusion gene analysis is a particular challenge in the clinical context due to the complex combination and numbers of driver genes and partner genes involved in chromosomal rearrangements [1, 11–14]. Moreover, the nucleic acid templates extracted from routine formalin fixed paraffin wax embedded tissues (FFPE) biopsy samples are characterised by low DNA/RNA yields with poor integrity and quality which is difficult to sequence particularly in relation to RNA fusion transcripts. Indeed, the challenges of genomic profiling of FFPE clinical samples was recently highlighted in the summary report of the Genomic England 100,000 genomes project which utilized fluorescent based sequencing and concluded that analysis of FFPE samples for personalised medicine was infeasible [15].

The increasing use of targeted agents offers the great advantage of increased specificity and reduced toxicity when compared with conventional chemotherapy [16]. Meta-analysis in diverse tumour types has shown that a personalized strategy of treatment is an independent predictor of better outcomes and fewer toxicity associated deaths when compared with chemotherapy [17]. The recent advances in targeted next-generation sequencing (NGS) technologies optimized for nucleic acid templates extracted from FFPE tumour samples now provides the opportunity to conduct precision oncology testing as part of the routine diagnostic work-flow. To investigate the potential role of clinically directed semiconductor sequencing in solid tumours we have established a clinically validated NGS platform optimised for analysis of FFPE clinical biopsy samples. This platform enables detection of 867 druggable driver-partner oncogenic fusions via analysis of 51 driver and 349 partner genes, with linkage to 140 targeted therapy protocols. All variants detected are "actionable" and therefore treatable by targeted therapies either on-market FDA and EMA approved, carrying ESMO and NCCN guideline references or currently in clinical trials, phases I-IV, worldwide [18].

Here we have undertaken a retrospective analysis of test trending data to investigate the types and frequency of clinically relevant fusions in solid tumours. Here we show that semiconductor sequencing can be incorporated into routine pathology diagnostic work-flows enabling detection of druggable fusions at high frequency across many solid tumour types.

## Materials and methods

### Patient demographics

A retrospective analysis was performed on the trending data generated as part of routine comprehensive precision oncology NGS testing for solid tumours and collected in compliance with ISO15189:2012 requirements for monitoring of quality indicators. The research conducted in this study was limited to secondary use of information previously collected in the course of normal care (without an intention to use it for research at the time of collection) and therefore does not require REC review. The patients and service users were not identifiable to

the research team carrying out trend data analysis. This is also in accordance with guidance issued by the National Research Ethics Service, Health Research Authority, NHS and follows the tenants of the Declaration of Helsinki. The trending data relates to a real-life cohort of 1112 patients tested between 14th February 2018 and 31st October 2019. The study cohort demographics are shown in S1 and S2 Tables. The cohort included all solid tumour types without application of inclusion or exclusion criteria and therefore representative of a larger population.

## Comprehensive NGS genomic profiling

The NGS platform utilized for clinical testing is validated for clinical use and accredited by CLIA (ID 99D2170813) and by UKAS (9376) in compliance with ISO15189:2012 and following the guidelines published by the Association for Molecular Pathology and College of American Pathologists and IQN-Path ASBL as described [8, 19]. The performance characteristics of the assay are shown in S3 Table. The NGS platform includes the targeting of 51 driver genes and 349 partner genes, enabling detection of 867 druggable driver-partner oncogenic fusions that is linked to 140 targeted therapy protocols. Genomic regions selected for analysis of clinically relevant fusions are shown in S4 Table.

## RNA extraction, library preparation and sequencing

RNA was extracted from FFPE curls cut at 10μm or from 5μm sections mounted onto unstained glass slides using the RecoverAll extraction kit (Ambion, Cat:A26069). RNA samples were diluted to 5ng/μl and reverse transcribed to cDNA in a 96 well plate using the Superscript Vilo cDNA synthesis kit (CAT 11754250). Library construction, template preparation, template enrichment and sequencing were performed using Ion Ampliseq library 2.0 (Cat: 4480441) and the Ion 540TM OT2 kit (Cat: A27753) according to the manufacturer's instructions. Sequencing was performed using the Ion S5 system 20 (Cat: A27212) utilising Ion 540TM chips (Cat:30 A27766).

## Quality control

Sequencing runs were quality controlled using the following parameters according to manufacturer's instructions (Ion Reporter™ 5.10.1.0): chip loading >60% with >45 million reads observed, enrichment 98–100%, polyclonal percentage <55%, low quality <26%, usable reads > 30% and aligned bases were ≥80%, unaligned bases were <20%, mean raw accuracy was >99% and overall read length between 100-115bp, average base coverage depth >1200, uniformity of amplicon (base) coverage >90%, amplicons were required to have less than 90% strand bias with >80% of amplicons reading end to end, on-target reads >85% and target base coverage at 1x, 20x, 100x and 500x >90% (S5 Table).

## Data analysis

Sequence alignment and variant calling was performed on The Torrent Suite™ Software (5.8.0). Alignment in Torrent Suite™ Software was performed using TMAP. The output BAM file was uploaded via the Ion Reporter Uploader plugin (5.8.32–1) to The Ion Reporter™ Software (5.10.1.0). Gene fusions were reported when occurring in >40 counts and meeting the threshold of assay specific internal RNA quality control with a sensitivity of 99% and PPV of 99%. Six internal expression quality controls were spiked into each sample to monitor assay performance with an acceptance cut-off of>15 reads in 5 out 6 controls [Ion Reporter™ 5.10.1.0; default fusion view 5.10] (S5 Table). The results of variant annotation were organized

hierarchically by gene, alteration, indication and level of evidence in relation to clinical action-ability following the joint recommendation of the association of the AMP/ASCO/CAP [8, 9]. Tertiary analysis software was used to link variants to curated lists of relevant labels, guidelines, and global clinical trials [Oncomine™ Reporter (Cat:A34298)]; GlobalData clinical trials database.

# Results

Eighty nine actionable fusion gene events were identified in 1112 samples of solid tumours (Fig 1 and S6 Table). Eighty two of the 1112 samples tested had at least one actionable fusion gene representing a frequency of 7.4% across the study cohort. The frequency of the different actionable gene fusions/rearrangements detected is shown in Fig 1. TBL1XR1-PIK3CA, MET-MET, WHSC1L1-FGFR1, FGFR3-TACC3 and EGFR-SEPT14 fusions and EGFR VIII rearrangements were identified as the most common druggable events (Fig 1). Seven of the samples harboured two fusion genes. Four of these seven cases relate to glioblastoma in which fusion pairs CAPZA2-MET and FIP1L1-PDGFRA, CAPZA2-MET and MET-MET, EGFR VIII and PTPRZ1-MET, PTPRZ1-MET and TBL1XR1-PIK3CA were identified. Fusion pairs were also identified in colon (CAPZA2-MET and MET-MET), lung (PIK3CA-TBL1XR1 and

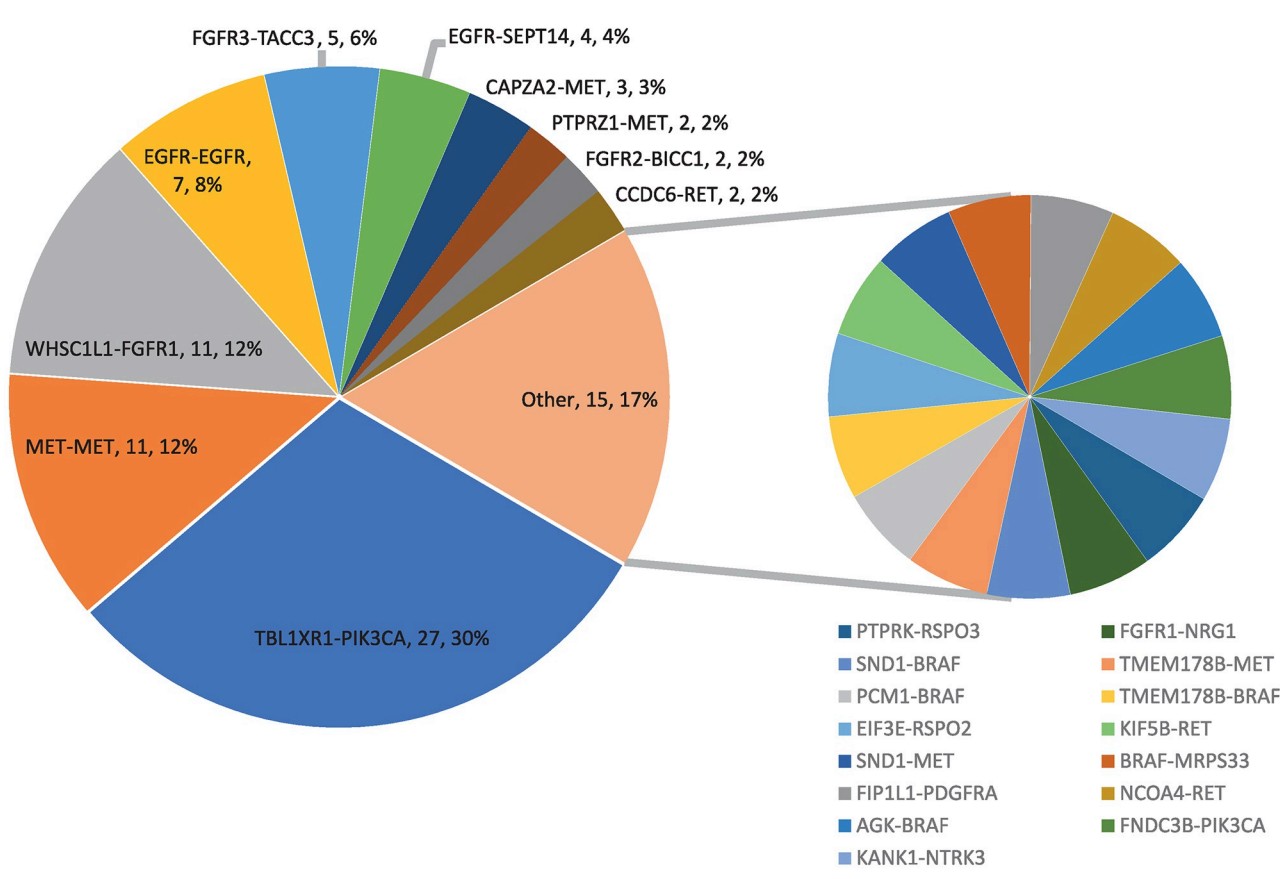

**Actionable Fusion Gene Landscape in Solid Tumours**

**Fig 1. Actionable fusion gene landscape in solid tumours.**

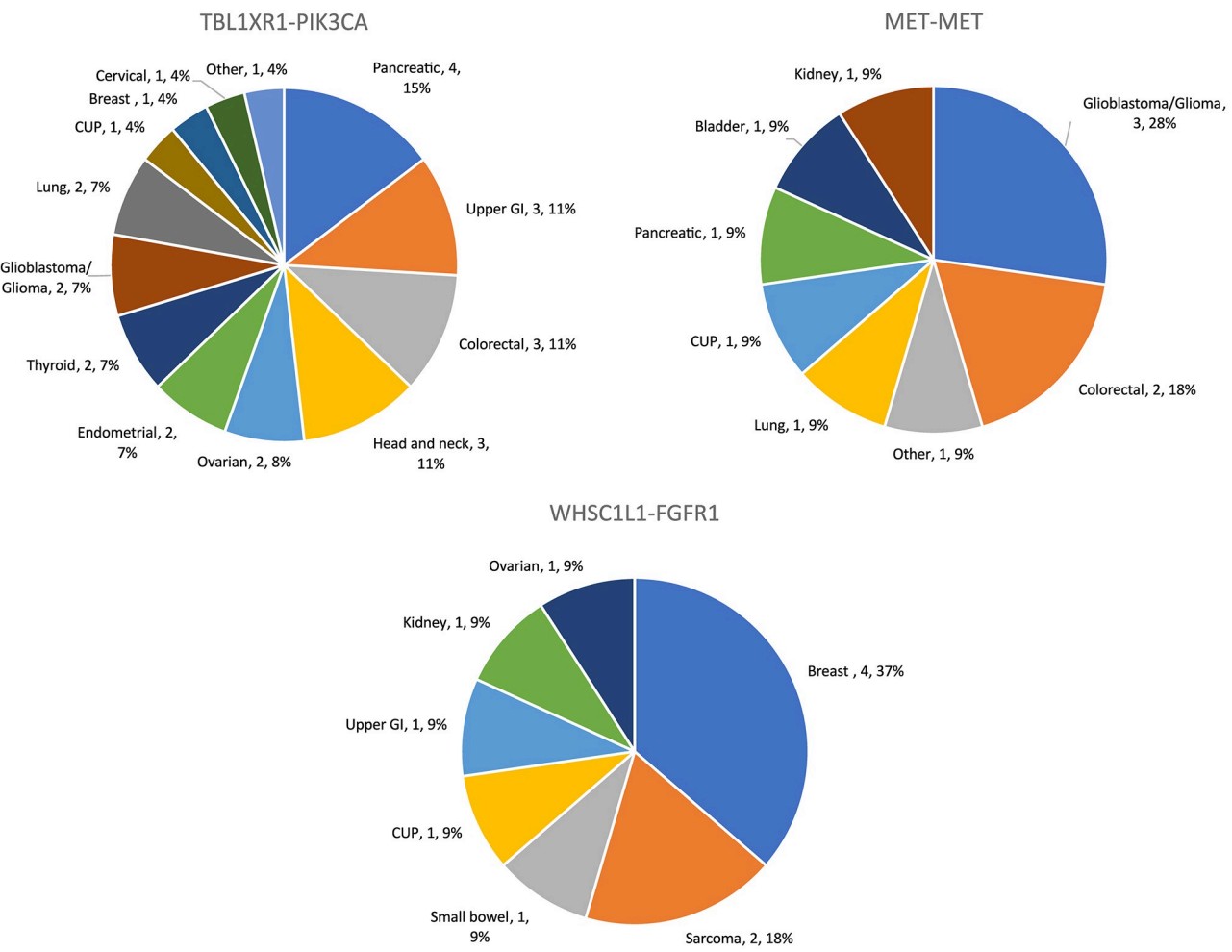

**Fig 2. Frequency of the three most common actionable gene fusions in solid tumours.**

RET-NCOA4) and pancreatic (FNDC3B-PIK3CA and TBL1XR1-PIK3CA) cancers (S6 Table).

The four most common fusions TBL1XR1-PIK3CA, MET-MET, WHSC1L1-FGFR1 and FGFR3-TACC3 representing 61% of all fusions were detected in a broad range of tumour types (Fig 2 and S6–S8 Tables). In contrast, some of the low frequency fusions, CCDC6-RET in thyroid and NSCLC lung cancer, FGFR2-BICC1 in cholangiocarcinoma, PTPRZ1-MET and KIF5B-RET in non-small cell lung (NSCLC) cancer (≤5 fusions detected) did show link-age to tumour type (Fig 1 and S6–S8 Tables). In tumour types with significant sample size (n >30), a high frequency (>7%) of actionable fusions was observed in glioblastoma, head and neck cancers, cancers of unknown primary (CUP), prostate and pancreatic cancers and in NSCLC (S8 Table). Glioblastoma, colorectal, lung and breast cancers harboured the most diverse set of actionable fusion genes (Fig 3). The fusions identified in the database were com-pared to those found in the TumorFusions data portal cataloguing over 20,000 gene fusions found in The Cancer Genome Atlas (TCGA) [20]. For gene fusions identified in both this study and the TCGA, there was little correspondence in the relative frequency of each fusion (S1 Fig). Fusions in the TCGA were also found in a variety of cancer types, for example the 36

FGFR3-TACC3 fusion events were spread across 10 different cancer types. No significant difference (p = 0.2) was found in the prevalence of fusions between primary (66/735, 9.0%) and metastatic (23/375, 6.1%) cancers and fusion read count showed no correlation with tumour percentage. In relation to read counts, liver cancer had a higher average when compared to colorectal cancer (p = 0.008), breast, pancreatic, sarcoma, endometrial, upper GI and thyroid cancers ($0.01 < p < 0.05$) using Tukey's multiple comparisons test on the geometric means (S2 Fig). However, most cancer types had too few fusions to make statistical comparisons.

Analysis of fusions in relation to age and gender are shown in S2 and S9 Tables. Although no associations were observed with age, WHSC1L1-FGFR1 appears to be more prevalent in females (p = 0.004) contrasting with FGFR3-TACC3 which is more prevalent in males (p = 0.01).

Actionable fusions can be targeted directly by small molecule inhibitors or alternatively through targeting of their cognate downstream signalling pathways. Bioinformatics pathway analysis showed the actionable fusions identified in this study were linked to a number of evolutionary conserved druggable cell signalling pathways including RAS/RAF/MEK/ERK, PIK3/AKT/MTOR, PLCγ/PKC, JAK/STAT and WNT/β catenin (Table 1). Detected fusions were bioinformatically linked to a total of 73 targeted therapy protocols. These included seven fusion-drug matches with FDA/EMA approval and/or NCCN/ESMO recommendations (Table 1) either in indication or approved in "cancer of other type" and therefore meeting the tier criteria I and II level of clinical significance as defined by the Joint Consensus Recommendation of the Association for Molecular Pathology, American Society of Clinical Oncology, and College of American Pathologists [8]. These fusions were additionally linked to a total of 80 clinical trials investigating the efficacy of drugs either targeting the kinase fusion directly or alternatively using inhibitors targeting pathways downstream (Table 1).

## Discussion

Advances in somatic cancer genetics and genomic profiling is leading to a shift in treatment paradigm from relatively non-specific empirically directed cytotoxic chemotherapies to a more biologically informed targeted approach [21–23]. The drug-target pairing that links a dysregulated molecular pathway with a cognate therapeutic agent defines the modern era of precision oncology. Targeted agents and immunotherapies are associated with superior response rates, fewer side effects and reduced healthcare costs when compared with nonselective chemotherapy [16, 17, 24].

Analysis of our NGS test trending data has shown that actionable fusions occur across a wide range of tumour types providing more personalised treatment options to cancer patients. Actionable fusions were identified at high frequency, 7.3% across all solid tumour types, rising to 23% in the case of glioblastoma. Notably, high frequency actionable fusions were agnostic of tumour type in keeping with previous findings, for example the TCGA RNA sequencing (RNA-seq) data corpus, in which high frequency fusions were observed across multiple tumour types [20]. In contrast, less prevalent actionable fusions exhibited tumour type specificity including CCDC6-RET fusions in thyroid and lung cancer, FGFR2-BICC1 in cholangiocarcinoma, PTPRZ1-MET in glioblastoma, EIF3E-RSPO2 and PTPRK-RSPO3 in colorectal cancer and KIF5B-RET in NSCLC as previously reported [3, 25–28]. Other low frequency actionable fusions namely, AGK-BRAF, FIP1L1-PDGFRA, FNDC3B-PIK3CA, RET-NCOA4, SND1-BRAF, TMEM178B-MET, TMEM178B-BRAF, KANK1-NTRK3, EIF3E-RSPO2 and PTPRK -RSPO3 have been previously reported as rare fusions in tumours of other type [1, 28–34]. Non targeted novel fusions were also identified including FGFR1-NRG1 in breast,

**Table 1.  Actionable fusion genes in solid tumours.**

| Genomic alteration | Cancer type | Pathway | Relevant therapies *approval in other cancer type | Clinical trials | Relevant Therapy |
|---|---|---|---|---|---|
| KANK1—NTRK3 | Breast | RAS/RAF/MEK/ERK | entrectinib | NCT02576431 | larotrectinib |
| | | PI3K/AKT/MTOR pathway | larotrectinib | NCT02637687 | larotrectinib |
| | | PLCγ/PKC | | NCT02568267 | entrectinib |
| | | | *Larotrectinib (all solid tumours) | NCT02465060 | larotrectinib |
| | | | | NCT03155620 | larotrectinib |
| | | | | NCT02920996 | merestinib |
| | | | | NCT03213704 | larotrectinib |
| | | | | NCT03297606 | temsirolimus |
| | | | | No NCT ID | entrectinib |
| | | | | NCT03093116 | repotrectinib |
| | | | | NCT03215511 | LOXO-195 |
| FGFR1—NRG1 | Breast | RAS/RAF/MEK/ERK | | NCT02052778 | TAS-120 |
| | | PI3K/AKT/MTOR pathway | | NCT02393248 | chemotherapy, INCMGA00012, pembrolizumab, pemigatinib, trastuzumab |
| | | PLCγ/PKC | | NCT01948297 | FF-284 |
| | | JAK/STAT | | NCT03834220 | FF-284 |
| | | | | NCT02272998 | ponatinib |
| | | | | NCT03297606 | sunitinib |
| | | | | NCT03160833 | HMPL-453 |
| | | | | NCT03929965 | anlotinib hydrochloride |
| | | | | NCT03235570 | pemigatinib |
| TMEM178B—MET | Colorectal | RAS/RAF/MEK/ERK | | NCT03297606 | crizotinib |
| | | PI3K/AKT | | NCT03175224 | bozitinib |
| | | | | NCT02219711 | sitravatinib |
| EGFR—SEPT14 | Colorectal, Glioblastoma | PI3K/AKT/MTOR pathway | | No NCT ID | cetuximab + chemotherapy |
| | | RAS/RAF/MEK/ERK | | No NCT ID | cetuximab + chemotherapy |
| | | | | NCT03152435 | CART-EGFR |
| | | | | NCT03454620 | GC1118A + chemotherapy |
| | | | | NCT03319459 | cetuximab + FATE-NK100 |
| | | | | NCT03297606 | erlotinib |
| | | | | NCT02013089 | erlotinib, gefitinib |
| | | | | NCT02423525 | afatinib |
| CAPZA2—MET | Glioblastoma, Colorectal | RAS/RAF/MEK/ERK | | NCT03598244 | volitinib |
| | | PI3K/AKT | | NCT03297606 | crizotinib |
| | | | | NCT03175224 | bozitinib |
| | | | | NCT02219711 | sitravatinib |
| FIP1L1—PDGFRA | Glioblastoma | RAS/RAF/MEK/ERK | | NCT03352427 | dasatinib, everolimus |
| | | PI3K/AKT/MTOR pathway | | | |

(Continued)

**Table 1.** (Continued)

| Genomic alteration | Cancer type | Pathway | *Relevant therapies* *approval in other cancer type* | *Clinical trials* | *Relevant Therapy* |
|---|---|---|---|---|---|
| *PTPRZ1—MET* | Glioblastoma | RAS/RAF/MEK/ERK | | NCT03598244 | volitinib |
| | | PI3K/AKT | | NCT03297606 | crizotinib |
| | | | | NCT03175224 | bozitinib |
| | | | | NCT02219711 | sitravatinib |
| *AGK—BRAF* | Glioblastoma | RAS/RAF/MEK/ERK | | NCT02639546 | cobimetinib |
| | | | | NCT02029001 | sorafenib |
| | | | | NCT03843775 | binimetinib + encorafenib |
| | | | | NCT03520075 | ASTX029 |
| | | | | NCT03905148 | lifirafenib, PD-0325901 |
| | | | | NCT03415126 | ASN007 |
| | | | | NCT02857270 | abemaciclib, cetuximab, chemotherapy, encorafenib, LY3214996, midazolam |
| | | | | NCT02607813 | LXH254 |
| | | | | NCT03634982 | RMC-4630 |
| *TBL1XR1—PIK3CA* | Breast, Cervical, Colorectal, CUP, endometrial, gastric, glioblastoma, head and neck, GOJ, NSCLC, Oesophageal, Ovarian, Pancreatic, Thyroid, Vulva | PI3K/AKT/MTOR pathway | | NCT03292250 | alpelisib |
| | | | | NCT03065062 | gedatolisib + palbociclib |
| | | | | NCT03297606 | temsirolimus |
| | | | | NCT02576444 | capivasertib, olaparib |
| | | | | NCT03673787 | atezolizumab + ipatasertib |
| | | | | NCT03805399 | chemotherapy, mTOR inhibitor |
| | | | | NCT02615730 | GSK-2636771 + chemotherapy |
| | | | | NCT03675893 | abemaciclib + LY-3023414 |
| *FGFR3—TACC3* | Glioblastoma, Head and Neck and prostate | RAS/RAF/MEK/ERK | *erdafitinib (Bladder cancer) | NCT03292250 | nintedanib |
| | | PI3K/AKT/MTOR pathway | | NCT01948297 | FF-284 |
| | | JAK/STAT | | NCT03834220 | FF-284 |
| | | | | NCT02272998 | ponatinib |
| | | | | NCT03297606 | sunitinib |
| | | | | NCT02052778 | TAS-120 |
| | | | | NCT03160833 | HMPL-453 |
| | | | | NCT03929965 | anlotinib hydrochloride |
| | | | | NCT03235570 | pemigatinib |
| *WHSC1L1—FGFR1* | Breast Cancer, CUP, Renal Cell Carcinoma, Oesophageal, Ovarian, Sarcoma, Small bowel | RAS/RAF/MEK/ERK | | NCT02872714 | pemigatinib |
| | | PI3K/AKT/MTOR pathway | | NCT02052778 | TAS-120 |
| | | JAK/STAT | | NCT03160833 | HMPL-453 |
| | | PLCy/PKC | | NCT03834220 | FF-284 |
| | | | | NCT02272998 | ponatinib |
| | | | | NCT03297606 | sunitinib |
| | | | | NCT01948297 | FF-284 |
| | | | | NCT03929965 | anlotinib hydrochloride |
| | | | | NCT03235570 | pemigatinib |
| | | | | NCT02393248 | chemotherapy, INCMGA00012, pembrolizumab, pemigatinib, trastuzumab |
| | | | | NCT03822117 | pemigatinib |
| | | | | NCT02052778 | futibatinib |

(Continued)

**Table 1.** (*Continued*)

| Genomic alteration | Cancer type | Pathway | *Relevant therapies* *approval in other cancer type* | *Clinical trials* | *Relevant Therapy* |
|---|---|---|---|---|---|
| *FGFR2— BICC1* | Cholangiocarcinoma (Liver) | RAS/RAF/ MEK/ERK | *erdafitinib (Bladder cancer) | NCT03230318 | derazantinib |
| | | PI3K/AKT/ MTOR pathway | | NCT03773302 | infigratinib |
| | | JAK/STAT | | NCT03656536 | pemigatinib |
| | | | | NCT02699606 | erdafitinib |
| | | | | NCT03834220 | FF-284 |
| | | | | NCT02150967 | infigratinib |
| | | | | NCT02924376 | pemigatinib |
| | | | | NCT02691793 | sunitinib |
| | | | | NCT02272998 | ponatinib |
| | | | | NCT03297606 | sunitinib |
| | | | | NCT02052778 | TAS-120 |
| | | | | NCT02393248 | chemotherapy, INCMGA00012, pembrolizumab, pemigatinib, trastuzumab |
| | | | | NCT03160833 | HMPL-453 |
| | | | | NCT01948297 | FF-284 |
| | | | | NCT03929965 | anlotinib hydrochloride |
| | | | | NCT03235570 | pemigatinib |
| *KIF5B—RET* | NSCLC | PI3K/AKT/ MTOR | Cabozantinib | No NCT ID | alectinib |
| | | RAS/RAF/ MEK/ERK | vandetanib | NCT03194893 | alectinib, crizotinib |
| | | PLCy/PKC | | NCT03391869 | ipilimumab, nivolumab, radiation therapy, surgical intervention |
| | | | | NCT01639508 | cabozantinib |
| | | | | NCT02314481 | alectinib |
| | | | | NCT03445000 | alectinib |
| | | | | NCT02540824 | apatinib |
| | | | | NCT02699606 | erdafitinib |
| | | | | NCT02299141 | nintedanib |
| | | | | NCT02664935 | sitravatinib |
| | | | | No NCT ID | targeted therapy, targeted therapy + chemotherapy |
| | | | | NCT03157128 | LOXO-292 |
| | | | | NCT03037385 | BLU-667 |
| | | | | NCT03780517 | BOS172738 |
| | | | | NCT02219711 | sitravatinib |
| | | | | NCT02029001 | sorafenib |
| | | | | NCT02450123 | sunitinib |
| | | | | NCT02691793 | sunitinib |
| | | | | NCT02272998 | ponatinib |
| | | | | NCT03297606 | sunitinib |
| *SND1—MET* | NSCLC | RAS/RAF/ MEK/ERK | | NCT03088930 | crizotinib |
| | | PI3K/AKT | | NCT02323126 | capmatinib + nivolumab |
| | | | | NCT02414139 | capmatinib |
| | | | | NCT02219711 | sitravatinib |
| | | | | NCT03297606 | crizotinib |
| | | | | NCT03175224 | bozitinib |

(*Continued*)

**Table 1.** (Continued)

| Genomic alteration | Cancer type | Pathway | Relevant therapies *approval in other cancer type | Clinical trials | Relevant Therapy |
|---|---|---|---|---|---|
| RET—NCOA4 | NSCLC | PI3K/AKT/MTOR | cabozantinib | No NCT ID | alectinib |
| | | RAS/RAF/MEK/ERK | vandetanib | NCT03194893 | alectinib, crizotinib |
| | | PLCy/PKC | | NCT03391869 | ipilimumab, nivolumab, radiation therapy, surgical intervention |
| | | | | NCT02314481 | alectinib |
| | | | | NCT03445000 | alectinib |
| | | | | NCT02540824 | apatinib |
| | | | | NCT01639508 | cabozantinib |
| | | | | NCT02699606 | erdafitinib |
| | | | | NCT02299141 | nintedanib |
| | | | | NCT02664935 | sitravatinib |
| | | | | No NCT ID | targeted therapy, targeted therapy + chemotherapy |
| | | | | NCT03157128 | LOXO-292 |
| | | | | NCT03037385 | BLU-667 |
| | | | | NCT03780517 | BOS172738 |
| | | | | NCT02219711 | sitravatinib |
| | | | | NCT02029001 | sorafenib |
| | | | | NCT02450123 | sunitinib |
| | | | | NCT02691793 | sunitinib |
| | | | | NCT02272998 | ponatinib |
| | | | | NCT03297606 | sunitinib |
| CCDC6—RET | NSCLC, Thyroid | PI3K/AKT/MTOR | cabozantinib (NSCLC) | No NCT ID | alectinib |
| | | RAS/RAF/MEK/ERK | vandetanib (NSCLC) | NCT03194893 | alectinib, crizotinib |
| | | PLCy/PKC | | NCT03391869 | ipilimumab, nivolumab, radiation therapy, surgical intervention |
| | | | | NCT02314481 | alectinib |
| | | | | NCT03445000 | alectinib |
| | | | | NCT02540824 | apatinib |
| | | | | NCT01639508 | cabozantinib |
| | | | | NCT02699606 | erdafitinib |
| | | | | NCT02299141 | nintedanib |
| | | | | NCT02664935 | sitravatinib |
| | | | | No NCT ID | targeted therapy, targeted therapy + chemotherapy |
| | | | | NCT03157128 | LOXO-292 |
| | | | | NCT03037385 | BLU-667 |
| | | | | NCT03780517 | BOS172738 |
| | | | | NCT02219711 | sitravatinib |
| | | | | NCT02029001 | sorafenib |
| | | | | NCT02450123 | sunitinib |
| | | | | NCT02691793 | sunitinib |
| | | | | NCT02272998 | ponatinib |
| | | | | NCT03297606 | sunitinib |
| | | | | NCT01945762 | vandetanib |

(Continued)

**Table 1.** (Continued)

| Genomic alteration | Cancer type | Pathway | *Relevant therapies* *approval in other cancer type* | *Clinical trials* | *Relevant Therapy* |
|---|---|---|---|---|---|
| MET—MET | Bladder, CUP, Colorectal, Glioblastoma, NSCLC, Pancreatic cancer, Squamous cell carcinoma, Renal cell carcinoma | RAS/RAF/MEK/ERK | crizotinib (NSCLC) | NCT03297606 | crizotinib |
| | | PI3K/AKT/MTOR | | NCT03175224 | bozitinib |
| | | | | NCT02219711 | sitravatinib |
| | | | | NCT03088930 | crizotinib |
| | | | | NCT03729297 | cabozantinib |
| | | | | NCT02323126 | capmatinib + nivolumab |
| | | | | NCT02213289 | rilotumumab + chemotherapy |
| | | | | NCT02414139 | capmatinib |
| | | | | NCT03598244 | volitinib |
| | | | | NCT02867592 | cabozantinib |
| | | | | NCT02465060 | crizotinib |
| | | | | NCT03911193 | cabozantinib |
| | | | | NCT02867592 | cabozantinib |
| | | | | NCT02465060 | crizotinib |
| TMEM178B—BRAF | Teratoma (Other) | RAS/RAF/MEK/ERK | | NCT02029001 | sorafenib |
| | | | | NCT03843775 | binimetinib + encorafenib |
| | | | | NCT03520075 | ASTX029 |
| | | | | NCT02639546 | cobimetinib |
| | | | | NCT03905148 | lifirafenib, PD-0325901 |
| | | | | NCT03415126 | ASN007 |
| | | | | NCT02857270 | abemaciclib, cetuximab, chemotherapy, encorafenib, LY3214996, midazolam |
| | | | | NCT02607813 | LXH254 |
| | | | | NCT03634982 | RMC-4630 |
| | | | | NCT02013089 | sorafenib, sunitinib |
| | | | | NCT03714958 | HDM-201 + trametinib |
| FNDC3B—PIK3CA | Pancreas | PI3K/AKT/MTOR pathway | | NCT03065062 | gedatolisib + palbociclib |
| | | | | NCT03297606 | temsirolimus |
| | | | | NCT02576444 | capivasertib, olaparib |
| | | | | NCT03673787 | atezolizumab + ipatasertib |
| SND1—BRAF | Prostate | RAS/RAF/MEK/ERK | | NCT02029001 | sorafenib |
| | | | | NCT03843775 | binimetinib + encorafenib |
| | | | | NCT03520075 | ASTX029 |
| | | | | NCT02639546 | cobimetinib |
| | | | | NCT03905148 | lifirafenib, PD-0325901 |
| | | | | NCT03415126 | ASN007 |
| | | | | NCT02857270 | abemaciclib, cetuximab, chemotherapy, encorafenib, LY3214996, midazolam |
| | | | | NCT02607813 | LXH254 |
| | | | | NCT03634982 | RMC-4630 |

(*Continued*)

**Table 1.** (Continued)

| Genomic alteration | Cancer type | Pathway | *approval in other cancer type | Clinical trials | Relevant Therapy |
|---|---|---|---|---|---|
| | | | *Relevant therapies* | | *Relevant Therapy* |
| BRAF—MRPS33 | Prostate | RAS/RAF/MEK/ERK | | NCT02029001 | sorafenib |
| | | | | NCT03843775 | binimetinib + encorafenib |
| | | | | NCT03520075 | ASTX029 |
| | | | | NCT02639546 | cobimetinib |
| | | | | NCT03905148 | lifirafenib, PD-0325901 |
| | | | | NCT03415126 | ASN007 |
| | | | | NCT02857270 | abemaciclib, cetuximab, chemotherapy, encorafenib, LY3214996, midazolam |
| | | | | NCT02607813 | LXH254 |
| | | | | NCT03634982 | RMC-4630 |
| EGFR—EGFR | Glioblastoma, Prostate | PI3K/AKT/MTOR pathway | | NCT03297606 | erlotinib |
| | | RAS/RAF/MEK/ERK | | NCT02423525 | afatinib |
| | | | | NCT03319459 | cetuximab + FATE-NK100 |
| PCM1—BRAF | Sarcoma | RAS/RAF/MEK/ERK | | NCT02639546 | cobimetinib |
| | | | | NCT02029001 | sorafenib |
| | | | | NCT03843775 | binimetinib + encorafenib |
| | | | | NCT03520075 | ASTX029 |
| | | | | NCT03905148 | lifirafenib, PD-0325901 |
| | | | | NCT03415126 | ASN007 |
| | | | | NCT02857270 | abemaciclib, cetuximab, chemotherapy, encorafenib, LY3214996, midazolam |
| | | | | NCT02607813 | LXH254 |
| | | | | NCT03634982 | RMC-4630 |
| PTPRK—RSPO3 | Colorectal | WNT/β catenin | | NCT01351103 | LGK974 |
| EIF3E—RSPO2 | Colorectal | WNT/β catenin | | NCT01351103 | LGK974 |

BRAF-MRPS33 in prostate, SND1-MET in lung, PCM1-BRAF in sarcoma and TMEM178B-MET in rectal cancer.

Detected actionable fusions were found to interact with one or more of the major evolutionary conserved cell signalling pathways, namely RAS/RAF/MEK/ERK, PI3K/AKT/MTOR, PLCy/PKC, JAK/STAT and WNT/β catenin. These are key signalling networks mediating fundamental processes including cell proliferation, differentiation, cell migration and apoptosis which are involved in homeostasis across all tissue types [35]. This may account for the fact that the most prevalent actionable fusions are independent of tumour type or tissue of origin. The minority population of tumour specific fusions identified also involve interaction with these conserved signalling networks suggesting that the tumour specificity of these particular fusions does not relate to tissue specific signalling pathways. Moreover, tissue specificity does not appear to be determined by tissue specific expression of the partner or driver genes involved in these fusions. For example, in the case of KIF5B-RET in NSCLC, KIF5B is expressed across a broad range of tissue types in keeping with its ubiquitous function as key component of the mitotic machinery and similarly RET is expressed across a broad range of tissue types [36, 37]. The mechanisms relating to tissue specificity remain poorly defined but may relate to other factors, for example interphase gene proximity (spatial proximity) that can

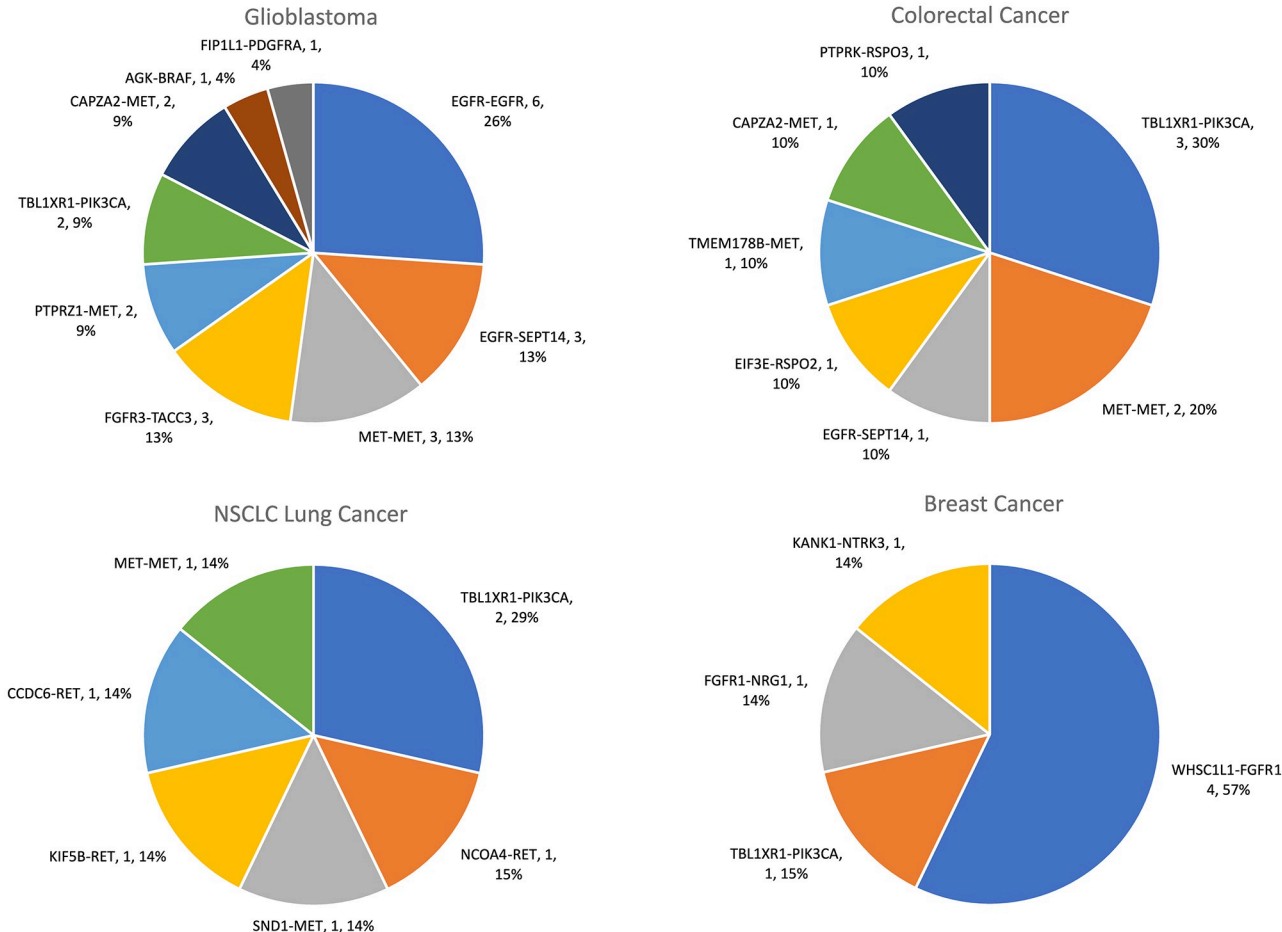

**Fig 3. Frequency of actionable gene fusions in glioblastoma, colorectal, NSCLC and breast cancers.**

facilitate generation of fusion genes [11, 38, 39]. Interestingly some fusions were associated with a particular tumour subtype, for example all fusions detected in breast cancer were restricted to cancers of ductal type and in lung cancers to those of NSCLC type. Cellular morphology has been postulated to represent a holistic readout of the complex genomic and gene expression changes in cancer cells and it is interesting to speculate that activation of the cognate signalling pathways linked to these particular fusions drive genomic changes and transcriptome profiles that act as determinants of these specific morpho-phenotypes [40].

The majority of detected druggable oncogenic driver genes represent tyrosine kinases (TKs) with partner genes encoding coil coiled domains leading to ligand-independent homo-dimerization, TK activation and dysregulated growth. However, it is also possible that disruption of the partner gene itself may also contribute to tumorigenesis through putative tumour suppressor roles. A number of the partner genes identified including TACC, CCDC6, EIF3E and KIF5B participate in DNA damage repair or mitotic chromosomal segregation [41–43]. Dysregulation of these genes have potential to drive error-prone DNA replication leading to genomic instability. This is in keeping with the notion that gene fusion events may function as a "two hit" model for multistep tumorigenesis.

Currently routine clinical molecular testing in relation to druggable fusion genes is limited to NSCLC, but even in this tumour type, analysis covers only a small number of potential

driver and partner genes, namely ALK and ROS rearrangements [10]. Here we have shown that semiconductor sequencing analysis of fusion transcripts in routine FFPE samples applied across all solid tumour types enables comprehensive analysis of hundreds of druggable fusion genes. Their detection enables clinical evidence-based linkage to a broad therapeutic armamentarium of targeted therapies which can have a major impact in the improved clinical management of advanced solid tumours. The majority of targeted agents are directed against the fusions themselves but alternatively treatment protocols also include targeting components of the cognate signalling pathways downstream. For example, the TBL1XR1-PIK3CA fusion is linked to targeted therapies inhibiting PI3K directly (alpelisib) but also inhibitors targeting downstream signalling components including AKT (capivasertib), MTOR (temsirolimus) and CDK4/6 (abemaciclib). Although a targeted agent for TRK fusions now has FDA approval, this rearrangement was identified as a rare event in our cohort of solid tumours (<1%) in keeping with previous reports [1]. In contrast here we have shown that targeting high frequency fusions such as TBL1XR1-PIK3CA, MET-MET, WHSC1L1-FGFR1 fusions offer much broader clinical utility across all tumour types.

In summary, we have shown that NGS semiconductor sequencing configured for detection of druggable fusions identifies a high frequency of actionable genetic rearrangements in solid tumours and that their coverage is therefore an important component of comprehensive precision oncology profiling.. Notably, high prevalence actionable fusions are not tumour type specific reinforcing the "site agnostic" approach to genomic profiling and supporting the concept of "molecular basket" clinical trials. Importantly we have also demonstrated that adoption of semiconductor sequencing methodologies enables comprehensive precision oncology profiling to be applied robustly to routine FFPE clinical biopsy samples allowing integration with globally established routine diagnostic pathology workflows.

## Ethical statement

The research conducted in this study was limited to secondary use of information previously collected in the course of normal care (without an intention to use it for research at the time of collection) and therefore does not require REC review. The patients and service users were not identifiable to the research team carrying out trend data analysis. This is also in accordance with guidance issued by the National Research Ethics Service, Health Research Authority, NHS and follows the tenants of the Declaration of Helsinki.

## Supporting information

**S1 Fig. Number of incidences of each fusion event as a proportion of the total considering only fusions found in both the TCGA database and the Oncologica dataset.**
(TIF)

**S2 Fig. Geometric mean of the number of fusions reads contained in each cancer type with 95% Wald confidence intervals.** ** represents p < 0.01 and * < 0.05 by Tukey's multipl comparisons test.
(TIF)

**S1 Table. Cancer type and histological classification of the study cohort.**
(PDF)

**S2 Table. Study cohort demographics by primary vs metastatic disease, age and gender.**
(PDF)

**S3 Table. Assay performance characteristics.**
(PDF)

**S4 Table. Sequences of the 867 driver partner fusion events targeted by the Oncofocus test.**
(PDF)

**S5 Table. Quality control metrics.**
(PDF)

**S6 Table. Detected actionable gene fusions in the sample cohort (n = 1,112).**
(PDF)

**S7 Table. Low frequency gene fusions detected in the sample cohort.**
(PDF)

**S8 Table. Frequency of actionable gene fusion and cancer type.**
(PDF)

**S9 Table. Detected driver genes and fusions by gender and age.**
(PDF)

## Acknowledgments

The authors thank the technical support staff at Oncologica UK Ltd for their valuable contribution to the study.

## Author Contributions

**Conceptualization:** Marco Loddo.

**Data curation:** Marco Loddo, Robert Thatcher.

**Formal analysis:** Keeda-Marie Hardisty, Alexander Llewelyn, Tiffany Haddow.

**Investigation:** Marco Loddo, Keeda-Marie Hardisty, Alexander Llewelyn, Gareth Williams.

**Methodology:** Keeda-Marie Hardisty, Tiffany Haddow, Robert Thatcher.

**Project administration:** Marco Loddo.

**Supervision:** Marco Loddo, Gareth Williams.

**Validation:** Keeda-Marie Hardisty.

**Visualization:** Alexander Llewelyn.

**Writing – original draft:** Marco Loddo, Gareth Williams.

**Writing – review & editing:** Marco Loddo, Gareth Williams.

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
