## [Decision Letter · Decision Letter 0]

11 Apr 2021

PONE-D-21-02472

Utilisation of semiconductor sequencing for detection of actionable fusions in solid tumors

PLOS ONE

Dear Dr. Loddo,

Thank you for submitting your manuscript to PLOS ONE. After careful consideration, we feel that it has merit but does not fully meet PLOS ONE’s publication criteria as it currently stands. Therefore, we invite you to submit a revised version of the manuscript that addresses the points raised during the review process.

We look forward to receiving your revised manuscript.

Kind regards,

Honey V. Reddi

Academic Editor

PLOS ONE

Journal Requirements:

2. In your Methods section, please provide additional information about the participant selection method and the demographic details of your participants. Please ensure you have provided sufficient details to replicate the analyses such as:

a) the date you accessed the patient data

b) a description of any inclusion/exclusion criteria that were applied to participant selection,

c) a table of relevant demographic details, and

d) a statement as to whether your sample can be considered representative of a larger population.

3, Thank you for stating the following financial disclosure:

N/A

3a)         Please clarify the sources of funding (financial or material support) for your study. List the grants or organizations that supported your study, including funding received from your institution.

3b)         State what role the funders took in the study. If the funders had no role in your study, please state: “The funders had no role in study design, data collection and analysis, decision to publish, or preparation of the manuscript.”

3c)          If any authors received a salary from any of your funders, please state which authors and which funders.

3d)         If you did not receive any funding for this study, please state: “The authors received no specific funding for this work.”

4. Thank you for providing the following Funding Statement: 

I have read the journal's policy and the authors of this manuscript have the following competing interests: Competing interests

ML and GW are shareholders and directors of Oncologica UK Ltd. ML, KH, TH, RT and GW are currently employed at Oncologica UK Ltd.

We note that one or more of the authors is affiliated with the funding organization, indicating the funder may have had some role in the design, data collection, analysis or preparation of your manuscript for publication; in other words, the funder played an indirect role through the participation of the co-authors.

If the funding organization did not play a role in the study design, data collection and analysis, decision to publish, or preparation of the manuscript and only provided financial support in the form of authors' salaries and/or research materials, please review your statements relating to the author contributions, and ensure you have specifically and accurately indicated the role(s) that these authors had in your study in the Author Contributions section of the online submission form. Please make any necessary amendments directly within this section of the online submission form.  Please also update your Funding Statement to include the following statement: “The funder provided support in the form of salaries for authors [insert relevant initials], but did not have any additional role in the study design, data collection and analysis, decision to publish, or preparation of the manuscript. The specific roles of these authors are articulated in the ‘author contributions’ section.”

If the funding organization did have an additional role, please state and explain that role within your Funding Statement.

Please also provide an updated Competing Interests Statement declaring this commercial affiliation along with any other relevant declarations relating to employment, consultancy, patents, products in development, or marketed products, etc. 

5a) If there are ethical or legal restrictions on sharing a de-identified data set, please explain them in detail (e.g., data contain potentially identifying or sensitive patient information) and who has imposed them (e.g., an ethics committee). Please also provide contact information for a data access committee, ethics committee, or other institutional body to which data requests may be sent.

5b) If there are no restrictions, please upload the minimal anonymized data set necessary to replicate your study findings as either Supporting Information files or to a stable, public repository and provide us with the relevant URLs, DOIs, or accession numbers. Please see http://www.bmj.com/content/340/bmj.c181.long for guidelines on how to de-identify and prepare clinical data for publication. For a list of acceptable repositories, please see http://journals.plos.org/plosone/s/data-availability#loc-recommended-repositories.

Additional Editor Comments:

The manuscript will need to revised in accordance with reviewer comments to be able to be considered for publication.

Reviewers' comments:

Reviewer's Responses to Questions

**Comments to the Author**

1. Is the manuscript technically sound, and do the data support the conclusions?

Reviewer #1: Partly

Reviewer #2: Yes

Reviewer #3: Yes

2. Has the statistical analysis been performed appropriately and rigorously? 

Reviewer #1: Yes

Reviewer #2: N/A

Reviewer #3: N/A

3. Have the authors made all data underlying the findings in their manuscript fully available?

Reviewer #1: Yes

Reviewer #2: Yes

Reviewer #3: Yes

4. Is the manuscript presented in an intelligible fashion and written in standard English?

Reviewer #1: Yes

Reviewer #2: Yes

Reviewer #3: Yes

5. Review Comments to the Author

Reviewer #1: This manuscript entitled “Utilisation of semiconductor sequencing for detection of actionable fusions in solid tumors” retrospectively evaluated 89 actionable fusions detected from 1112 patients with solid tumors. The authors analyzed genomic profiling, targeted therapies, and clinical trials of the identified fusions. This manuscript is well written and below are the comments.

Major comments:

1. The authors conclude that the majority of detected actionable fusions are independent of tumor type or tissue of origin. However, Table 1 and Figure 2 shows that there are specific fusions only detected in specific tumor types, such as EGFR VIII in glioma and prostate cancer. In addition, other retrospective studies have concluded that the frequency of fusions identified is dependent on specific tumor type (e.g. Pavalan et al., 2019). The authors should provide other data/evidences to support this conclusion.

2. In addition, this study identified 51 driver genes in the driver-partner fusions. It would be more comprehensive if the authors could include additional analysis of actionable variants of these driver genes to further conclude the association between actionable fusions and the tumor types.

3. Figure 1 and Figure 2 provide frequency of the actionable gene fusions in solid tumors, is it possible to provide a summary table of the molecular findings separated by age and gender?

4. The manuscript only includes the 89 actionable fusion gene identified. Can the author also provide a summary table of the other fusions detected in different tumor types?

Minor comments:

1. Second and third paragraphs of the discussion sections can be moved to the result section.

2. Type and writing errors in introduction and discussion section (e.g. line 226, 228)

3. There are many fusion databases available (e.g. TCGA fusion), the authors should consider comparing the fusion distribution from this study to the databases.

Reviewer #2: This is an interesting study to demonstrate the frequency of fusions in solid tumors and the importance and opportunity for expanded precision medicine in patient treatment. The manuscript is well written and does provide a good basis of information however it can be expanded further for impact and significance.

Specific comments:

1. Expansion on the cohort:

a. Overall information for whole group such as, tumors originated in females or males, age demarcation, tumor type is primary or metastatic, etc.

b. Further cohort delineation in regard to whether the cohort was screened for other immunohistochemical status or molecular testing such as SNVs or MNVs. Are the identified fusions seen in SNV negative samples or equally? Etc.

c. Outcome information for any potential correlation to fusion containing tumors and disease outcome

2. Methods section clarifications are needed to ensure the ability to replicate the data. In addition, in supplementary tables 2 and 3, Oncofocus test is mentioned but not described at all in the methods or other sections.

3. Fusion read count could be expanded, and any correlation between tumor types, tumor percentage and fusion read count could be valuable.

4. The figures focus on frequency however adding the number of cases that had the specific fusions to the charts would add to the figures.

Reviewer #3: 4/3/2021

Dear Dr., Reddi,

Thank you for inviting me to review the original research article titled “Utilisation of semiconductor sequencing for detection of actionable fusions in solid tumors” by Loddo et al.

This is a well-written manuscript emphasizing the importance of fusion gene analysis and precision medicine in the NGS era. Especially, the discussion section of the manuscript covered and linked an array of interesting topics. I do not see any major flaws in the manuscript, and I recommend that this manuscript can be accepted for publication once addressing the following suggestions.

Comments

1) Line 34 has a typo with an extra “W” before the word “Where”.

2) In Line 65, the sentence starting with “This platform enables..” can be rephrased as “This platform enables detection of 867 druggable driver-partner oncogenic fusions via analysis of 51 driver and 349 partner genes, with linkage to 140 targeted 68 therapy protocols”.

3) Line 96 can be rephrased as “ The NGS platform includes the targeting of 51 driver genes and 349 partner genes enabling detection of 867 druggable driver-partner oncogenic fusions that is linked to 140 targeted therapy protocols.”

4) In Lines 137 and 141 – I would not call EGFRvIII as a gene fusion event, but it is a gene rearrangement.

5) In Line 228 – there is a typo “ keeping wFith”

6) Line 231 – The first sentence can be rephrased.

Thanks,

Pavalan

6. PLOS authors have the option to publish the peer review history of their article (what does this mean?). If published, this will include your full peer review and any attached files.

Reviewer #1: No

Reviewer #2: No

Reviewer #3: No

---

## [Author Response · Author response to Decision Letter 0]

27 Dec 2021

Dear Dr Reddi, 

Many thanks for your email. We have now addressed your requests and wish to submit the revised manuscript. 

We look forward to your reply.

Best Wishes

Dr Marco Loddo

BSc PhD

Professor Gareth H Williams

BSc MBChB PhD FRCPath FLSW

PONE-D-21-02472R1

Utilisation of semiconductor sequencing for detection of actionable fusions in solid tumors Dr Marco Loddo

Dear Dr. Loddo,

We've checked your submission and before we can proceed, we need you to address the following issues:

1, Thank you for stating the following financial disclosure:

N/A

1a) Please clarify the sources of funding (financial or material support) for your study. List the grants or organizations that supported your study, including funding received from your institution.

1b) State what role the funders took in the study. If the funders had no role in your study, please state: “The funders had no role in study design, data collection and analysis, decision to publish, or preparation of the manuscript.”

1c) If any authors received a salary from any of your funders, please state which authors and which funders.

1d) If you did not receive any funding for this study, please state: “The authors received no specific funding for this work.”

We have included the following paragraph at the end of the manuscript:

“The authors received no specific funding for this work. Data analysis conducted in this study was limited to secondary use of information previously collected in the course of normal care. No dedicated funding source was allocated for this study. The funder provided support in the form of salaries for all authors, but did not have any additional role in the study design, data collection and analysis, decision to publish, or preparation of the manuscript. The specific roles of these authors are articulated in the ‘author contributions’ section.” Pages 19-20, lines 272-277.

2. Thank you for providing the following Funding Statement:

I have read the journal's policy and the authors of this manuscript have the following competing interests: Competing interests

ML and GW are shareholders and directors of Oncologica UK Ltd. ML, KH, TH, RT and GW are currently employed at Oncologica UK Ltd.

We note that one or more of the authors is affiliated with the funding organization, indicating the funder may have had some role in the design, data collection, analysis or preparation of your manuscript for publication; in other words, the funder played an indirect role through the participation of the co-authors.

If the funding organization did not play a role in the study design, data collection and analysis, decision to publish, or preparation of the manuscript and only provided financial support in the form of authors' salaries and/or research materials, please review your statements relating to the author contributions, and ensure you have specifically and accurately indicated the role(s) that these authors had in your study in the Author Contributions section of the online submission form. Please make any necessary amendments directly within this section of the online submission form. Please also update your Funding Statement to include the following statement: “The funder provided support in the form of salaries for authors [insert relevant initials], but did not have any additional role in the study design, data collection and analysis, decision to publish, or preparation of the manuscript. The specific roles of these authors are articulated in the ‘author contributions’

section.”

If the funding organization did have an additional role, please state and explain that role within your Funding Statement.

Please also provide an updated Competing Interests Statement declaring this commercial affiliation along with any other relevant declarations relating to employment, consultancy, patents, products in development, or marketed products, etc.

We have clarified these points in the above paragraph which has been inserted at the end of the manuscript. Pages 19-20, lines 272-277.

3. We note that your Data Availability statement states the following: "The datasets analysed during the current study are provided in the Supplementary File. Further technical details are available from the corresponding author upon reasonable request."

PLOS journals require that all data presented in the study be made publicly available at or before the time of publication. If there are legal or ethical restrictions on the data being made publicly available, such as IRB restriction or patient confidentiality, authors must provide a way for fellow researchers to access the data.

At this time, please confirm that your submission contains your "minimal data set", which PLOS defines as consisting of the data set used to reach the conclusions drawn in the manuscript with related metadata and methods, and any additional data required to replicate the reported study findings in their entirety. This includes:

We can confirm that the submission contains the minimal dataset.

Values reported in Results for fusion frequencies, metastatic status and fusion read counts (page 7) derived from raw data in Supplementary Table 6, summarised for each cancer type in Supplementary Table 8. Age and gender data (page 8) is located in Supplementary Tables 2 and 9. Table 1 contains data used to report targeted therapy protocols (page 8).

Figures 1, 2 and 3 contain raw values in legend, with data derived from Supplementary tables 6 and 8.

3) The points extracted from images for analysis.

N/A

---

## [Decision Letter · Decision Letter 1]

8 Feb 2022

PONE-D-21-02472R1Utilisation of semiconductor sequencing for detection of actionable fusions in solid tumorsPLOS ONE

Dear Dr. Loddo

Thank you for submitting your manuscript to PLOS ONE. After careful consideration, we feel that it has merit but does not fully meet PLOS ONE’s publication criteria as it currently stands. Therefore, we invite you to submit a revised version of the manuscript that addresses the points raised during the review process.

We look forward to receiving your revised manuscript.

Kind regards,

Honey V. Reddi

Academic Editor

PLOS ONE

Additional Editor Comments (if provided):

Dear Dr. Loddo - While some of your responses have been accepted by our reviewers,  reviewer 1 has noted that you have not provided a specific response to their earlier comments and it is important that those comments be addressed.

Reviewers' comments:

Reviewer's Responses to Questions

**Comments to the Author**

1. If the authors have adequately addressed your comments raised in a previous round of review and you feel that this manuscript is now acceptable for publication, you may indicate that here to bypass the “Comments to the Author” section, enter your conflict of interest statement in the “Confidential to Editor” section, and submit your "Accept" recommendation.

Reviewer #1: (No Response)

Reviewer #3: All comments have been addressed

2. Is the manuscript technically sound, and do the data support the conclusions?

Reviewer #1: Partly

Reviewer #3: Yes

3. Has the statistical analysis been performed appropriately and rigorously? 

Reviewer #1: Yes

Reviewer #3: Yes

4. Have the authors made all data underlying the findings in their manuscript fully available?

Reviewer #1: Yes

Reviewer #3: Yes

5. Is the manuscript presented in an intelligible fashion and written in standard English?

Reviewer #1: Yes

Reviewer #3: Yes

6. Review Comments to the Author

Reviewer #1: The authors did not response to the comments below.

This manuscript entitled “Utilisation of semiconductor sequencing for detection of actionable fusions in solid tumors” retrospectively evaluated 89 actionable fusions detected from 1112 patients with solid tumors. The authors analyzed genomic profiling, targeted therapies, and clinical trials of the identified fusions. This manuscript is well written and below are the comments.

Major comments:

1. The authors conclude that the majority of detected actionable fusions are independent of tumor type or tissue of origin. However, Table 1 and Figure 2 shows that there are specific fusions only detected in specific tumor types, such as EGFR VIII in glioma and prostate cancer. In addition, other retrospective studies have concluded that the frequency of fusions identified is dependent on specific tumor type (e.g. Pavalan et al., 2019). The authors should provide other data/evidences to support this conclusion.

2. In addition, this study identified 51 driver genes in the driver-partner fusions. It would be more comprehensive if the authors could include additional analysis of actionable variants of these driver genes to further conclude the association between actionable fusions and the tumor types.

3. Figure 1 and Figure 2 provide frequency of the actionable gene fusions in solid tumors, is it possible to provide a summary table of the molecular findings separated by age and gender?

4. The manuscript only includes the 89 actionable fusion gene identified. Can the author also provide a summary table of the other fusions detected in different tumor types?

Minor comments:

1. Second and third paragraphs of the discussion sections can be moved to the result section.

2. Type and writing errors in introduction and discussion section (e.g. line 226, 228)

3. There are many fusion databases available (e.g. TCGA fusion), the authors should consider comparing the fusion distribution from this study to the databases.

Reviewer #3: Thank you authors for your efforts to include my suggestions in the manuscript. I am certain that this manuscript is acceptable for publication.

7. PLOS authors have the option to publish the peer review history of their article (what does this mean?). If published, this will include your full peer review and any attached files.

Reviewer #1: No

Reviewer #3: No

---

## [Author Response · Author response to Decision Letter 1]

17 May 2022

23 August 2021

Dear Dr Reddi, 

Many thanks for your email. We have now address the referee’s comments and wish to submit the revised manuscript. 

Please find below a point by point response to the referee’s comments as requested. 

We look forward to your reply.

Best Wishes

Professor Gareth H Williams

BSc MBChB PhD FRCPath FLSW

 Dr Marco Loddo

BSc PhD

2. In your Methods section, please provide additional information about the participant selection method and the demographic details of your participants. Please ensure you have provided sufficient details to replicate the analyses such as:

a) the date you accessed the patient data. This is recorded in the material and methods section, page 4, lines 88-89. We have now added the precise dates to further clarify the temporal aspects of the cohort. 

b) a description of any inclusion/exclusion criteria that were applied to participant selection. No inclusion or exclusion criteria were applied, analysis was conducted on all tumour types tested between the specified dates

c) a table of relevant demographic details. A sentence has been inserted in the materials and methods section, page 4, line 89 “The demographics of the cohort are summarized in Supplementary Table 1 and 2”. 

d) a statement as to whether your sample can be considered representative of a larger population.

The cohort included all solid tumour types without application of inclusion or exclusion criteria and therefore representative of a larger population. This sentence has been included in the revised manuscript, pages 4, 5, lines 89-91.

Reviewer #1: This manuscript entitled “Utilisation of semiconductor sequencing for detection of actionable fusions in solid tumors” retrospectively evaluated 89 actionable fusions detected from 1112 patients with solid tumors. The authors analyzed genomic profiling, targeted therapies, and clinical trials of the identified fusions. This manuscript is well written and below are the comments.

We are grateful for the supportive comments on our manuscript.

Major comments:

1. The authors conclude that the majority of detected actionable fusions are independent of tumor type or tissue of origin. However, Table 1 and Figure 2 shows that there are specific fusions only detected in specific tumor types, such as EGFR VIII in glioma and prostate cancer.

As highlighted in the results section page 7 lines 140-148 the high prevalence fusions TBL1XR1-PIK3CA, MET-MET and WHSC1L1-FGFR1 are independent of tumour types whereas the lower prevalence fusions (around the 6% cut point) are restricted to certain tumour types in keeping with previous studies (Table 1, Figure 2 and Supplementary Table 6). The referee is entirely correct in identifying the confusing sentence “the majority of detected actionable fusions are independent of tumour type” which is present in the abstract (page 2, line 22) and last paragraph of the discussion (page 18, line 256) and grateful for picking up this critical point. These three genes constitute the “majority” of actionable fusion events but represent only a minority (not majority) of all fusion types detected. Once the prevalence reaches around 6% the fusions start to show tumour specificity. The author is also correct that EGFR VIII is showing tumour specificity in relation to glioma and prostate cancer and should not be included in Figure 2 but Supplementary Table 6. This has been amended. Moreover FGFR3-TACC3 which has been included in supplementary table 6 actually shows distribution across multiple tumour types including Head & Neck, prostate and glial tumours. To address these critical points we have made the following changes to the manuscript

1) We have moved EGFR VIII to Supplementary Table 7

2) We have removed EGFR VIII from Figure 2. We have replaced EGFR VIII with FGFR3-TACC3 in the result section, page 7, lines 152- 154

3) We have corrected the sentences which include “majority of fusions ….. with “more prevalent/most prevalent, pages 2, line 22; page 17, line 202; page 17, 218; page 19, line 261 

 In addition, other retrospective studies have concluded that the frequency of fusions identified is dependent on specific tumor type (e.g. Pavalan et al., 2019). The authors should provide other data/evidences to support this conclusion.

The pavalan et al., 2019 study was a screen of all fusions including actionable, unknown significance and benign. Only 7 were classified as actionable over a cohort of 183 tumour samples and therefore it is difficult to compare the data with our study which focuses exclusively on actionable fusions

1) We cite the TCGA RNA sequencing (RNA-seq) data corpus to support our findings that high prevalence fusions are agnostic of tumour type for example the 36 FGFR3-TACC3 fusion events were spread across 10 different cancer types, page 8, lines 160-163. 

2) The results and discussion section cites publications relating to tumour specific fusions in keeping with our results. These are referenced in the paper (24-35)

2. In addition, this study identified 51 driver genes in the driver-partner fusions. It would be more comprehensive if the authors could include additional analysis of actionable variants of these driver genes to further conclude the association between actionable fusions and the tumour types.

The assay platform targets 505 genes and detects actionable genetic variants linked to 780 anti-cancer targeted therapies/therapy combinations. This includes the 51 driver genes relating to the fusions analysed in this study. We are presently analysing all variants CNVs, deletions, indels, SNVs etc across the cohort. This is a complex pan-cancer big data study is beyond the scope of the present manuscript. We will however be citing the data relating to fusions in the forthcoming manuscript and collating with other somatic variants including mutually exclusive or concurrent genetic alterations to determine pathway interactions. 

3. Figure 1 and Figure 2 provide frequency of the actionable gene fusions in solid tumours, is it possible to provide a summary table of the molecular findings separated by age and gender?

We have now conducted this analysis and the following data inserted into the manuscript. 

Analysis of fusions in relation to age and gender are shown in Supplementary Table 2 and 9. Although no associations were observed with age, WHSC1L1-FGFR1 appears to be more prevalent in females contrasting with FGFR3-TACC3 which is more prevalent in males (page 8, lines 170-172; supplementary table 2 and 9)

4. The manuscript only includes the 89 actionable fusion gene identified. Can the author also provide a summary table of the other fusions detected in different tumour types?

The linkage between fusion type and tumour type are summarised in Supplementary Tables 6-8. This is a targeted NGS assay designed to detect 867 druggable driver-partner oncogenic fusions via analysis of 51 driver and 349 partner genes, with linkage to 140 targeted therapy protocols. This is not a screening assay and therefore non actionable fusions e.g. unknown significance, benign were not identified, page 5, line 101-103. 

Minor comments:

1. Second and third paragraphs of the discussion sections can be moved to the result section.

With insertion of additional content in response to referees comments we feel on balance that these paragraphs are best placed in the discussion

2. Type and writing errors in introduction and discussion section (e.g. line 226, 228)

Thank you for highlighting these errors. These have now been amended.

3. There are many fusion databases available (e.g. TCGA fusion), the authors should consider comparing the fusion distribution from this study to the databases.

We have now compared our data to fusion data databases as requested including correlation of fusion read counts with tumour type and percentage. This has been incorporated into the results section and discussion section, page 8, lines 167-171 and page 17, lines 202- 204.

Reviewer #2: This is an interesting study to demonstrate the frequency of fusions in solid tumors and the importance and opportunity for expanded precision medicine in patient treatment. The manuscript is well written and does provide a good basis of information however it can be expanded further for impact and significance.

We are most grateful for the supportive comments on our manuscript.

Specific comments:

1. Expansion on the cohort:

a. Overall information for whole group such as, tumors originated in females or males, age demarcation, tumor type is primary or metastatic, etc.

We agree these are important associations and have conducted the appropriate analysis. This data has now been incorporated into Supplementary Table 2 and results section, page 8, lines 172-174.

b. Further cohort delineation in regard to whether the cohort was screened for other immunohistochemical status or molecular testing such as SNVs or MNVs. Are the identified fusions seen in SNV negative samples or equally? Etc.

These are interesting questions including whether loss of DDR function and associated genomic instability might show linkage with fusions. The assay platform targets 505 genes and detects actionable genetic variants linked to 780 anti-cancer targeted therapies/therapy combinations. This includes the 51 driver genes relating to the fusions analysed in this study. We are presently analysing all variants CNVs, deletions, indels, SNVs etc across the cohort. This is a complex pan-cancer big data study which is beyond the scope of the present manuscript. We will however be citing the data relating to fusions in the forthcoming manuscript and collating with other somatic variants including mutually exclusive or concurrent genetic alterations to determine pathway interactions.

c. Outcome information for any potential correlation to fusion containing tumours and disease outcome

This study is a retrospective analysis of test trending data relating to our clinical service. Unfortunately our data is restricted to pathological information only and does not include outcome data. 

2. Methods section clarifications are needed to ensure the ability to replicate the data. In addition, in supplementary tables 2 and 3, Oncofocus test is mentioned but not described at all in the methods or other sections.

Oncofocus was mentioned in error as it refers to the brand name of the assay. This has now been corrected. 

3. Fusion read count could be expanded, and any correlation between tumor types, tumor percentage and fusion read count could be valuable.

Fusion read count in relation to tumour type and percentage has now been incorporated into the manuscript, page 8, lines 167-171.

4. The figures focus on frequency however adding the number of cases that had the specific fusions to the charts would add to the figures.

Case numbers have now been included in the Figures. 

Reviewer #3: 4/3/2021

Dear Dr., Reddi,

Thank you for inviting me to review the original research article titled “Utilisation of semiconductor sequencing for detection of actionable fusions in solid tumors” by Loddo et al.

This is a well-written manuscript emphasizing the importance of fusion gene analysis and precision medicine in the NGS era. Especially, the discussion section of the manuscript covered and linked an array of interesting topics. I do not see any major flaws in the manuscript, and I recommend that this manuscript can be accepted for publication once addressing the following suggestions.

Comments

1) Line 34 has a typo with an extra “W” before the word “Where”.

This has been corrected. 

2) In Line 65, the sentence starting with “This platform enables..” can be rephrased as “This platform enables detection of 867 druggable driver-partner oncogenic fusions via analysis of 51 driver and 349 partner genes, with linkage to 140 targeted 68 therapy protocols”.

We have inserted the suggested sentence, page 4, line 68-70.

3) Line 96 can be rephrased as “ The NGS platform includes the targeting of 51 driver genes and 349 partner genes enabling detection of 867 druggable driver-partner oncogenic fusions that is linked to 140 targeted therapy protocols.”

We have inserted the suggested sentence, page 5, line 101-103.

4) In Lines 137 and 141 – I would not call EGFRvIII as a gene fusion event, but it is a gene rearrangement.

We have replaced EGFR VIII fusion with rearrangement, page 7, line 141, 142.

5) In Line 228 – there is a typo “ keeping wFith”

This has been corrected. 

6) Line 231 – The first sentence can be rephrased.

This sentence has been rephrased. 

 23 August 2021

Dear Dr Reddi, 

Many thanks for your email. We have now addressed your requests and wish to submit the revised manuscript. 

We look forward to your reply.

Best Wishes

Dr Marco Loddo

BSc PhD

Professor Gareth H Williams

BSc MBChB PhD FRCPath FLSW

PONE-D-21-02472R1

Utilisation of semiconductor sequencing for detection of actionable fusions in solid tumors Dr Marco Loddo

Dear Dr. Loddo,

We've checked your submission and before we can proceed, we need you to address the following issues:

1, Thank you for stating the following financial disclosure:

N/A

1a) Please clarify the sources of funding (financial or material support) for your study. List the grants or organizations that supported your study, including funding received from your institution.

1b) State what role the funders took in the study. If the funders had no role in your study, please state: “The funders had no role in study design, data collection and analysis, decision to publish, or preparation of the manuscript.”

1c) If any authors received a salary from any of your funders, please state which authors and which funders.

1d) If you did not receive any funding for this study, please state: “The authors received no specific funding for this work.”

We have included the following paragraph at the end of the manuscript:

“The authors received no specific funding for this work. Data analysis conducted in this study was limited to secondary use of information previously collected in the course of normal care. No dedicated funding source was allocated for this study. The funder provided support in the form of salaries for all authors, but did not have any additional role in the study design, data collection and analysis, decision to publish, or preparation of the manuscript. The specific roles of these authors are articulated in the ‘author contributions’ section.” Pages 19-20, lines 272-277.

2. Thank you for providing the following Funding Statement:

I have read the journal's policy and the authors of this manuscript have the following competing interests: Competing interests

ML and GW are shareholders and directors of Oncologica UK Ltd. ML, KH, TH, RT and GW are currently employed at Oncologica UK Ltd.

We note that one or more of the authors is affiliated with the funding organization, indicating the funder may have had some role in the design, data collection, analysis or preparation of your manuscript for publication; in other words, the funder played an indirect role through the participation of the co-authors.

If the funding organization did not play a role in the study design, data collection and analysis, decision to publish, or preparation of the manuscript and only provided financial support in the form of authors' salaries and/or research materials, please review your statements relating to the author contributions, and ensure you have specifically and accurately indicated the role(s) that these authors had in your study in the Author Contributions section of the online submission form. Please make any necessary amendments directly within this section of the online submission form. Please also update your Funding Statement to include the following statement: “The funder provided support in the form of salaries for authors [insert relevant initials], but did not have any additional role in the study design, data collection and analysis, decision to publish, or preparation of the manuscript. The specific roles of these authors are articulated in the ‘author contributions’

section.”

If the funding organization did have an additional role, please state and explain that role within your Funding Statement.

Please also provide an updated Competing Interests Statement declaring this commercial affiliation along with any other relevant declarations relating to employment, consultancy, patents, products in development, or marketed products, etc.

We have clarified these points in the above paragraph which has been inserted at the end of the manuscript. Pages 19-20, lines 272-277.

3. We note that your Data Availability statement states the following: "The datasets analysed during the current study are provided in the Supplementary File. Further technical details are available from the corresponding author upon reasonable request."

PLOS journals require that all data presented in the study be made publicly available at or before the time of publication. If there are legal or ethical restrictions on the data being made publicly available, such as IRB restriction or patient confidentiality, authors must provide a way for fellow researchers to access the data.

At this time, please confirm that your submission contains your "minimal data set", which PLOS defines as consisting of the data set used to reach the conclusions drawn in the manuscript with related metadata and methods, and any additional data required to replicate the reported study findings in their entirety. This includes:

We can confirm that the submission contains the minimal dataset.

Values reported in Results for fusion frequencies, metastatic status and fusion read counts (page 7) derived from raw data in Supplementary Table 6, summarised for each cancer type in Supplementary Table 8. Age and gender data (page 8) is located in Supplementary Tables 2 and 9. Table 1 contains data used to report targeted therapy protocols (page 8).

Figures 1, 2 and 3 contain raw values in legend, with data derived from Supplementary tables 6 and 8.

3) The points extracted from images for analysis.

N/A

 23 August 2021

Dear Dr Reddi, 

Many thanks for your email. We have now addressed your requests and wish to submit the revised manuscript. 

We look forward to your reply.

Best Wishes

Dr Marco Loddo

BSc PhD

Professor Gareth H Williams

BSc MBChB PhD FRCPath FLSW

PONE-D-21-02472R1

Utilisation of semiconductor sequencing for detection of actionable fusions in solid tumors Dr Marco Loddo

Dear Dr. Loddo,

We've checked your submission and before we can proceed, we need you to address the following issues:

1, Thank you for stating the following financial disclosure:

N/A

1a) Please clarify the sources of funding (financial or material support) for your study. List the grants or organizations that supported your study, including funding received from your institution.

1b) State what role the funders took in the study. If the funders had no role in your study, please state: “The funders had no role in study design, data collection and analysis, decision to publish, or preparation of the manuscript.”

1c) If any authors received a salary from any of your funders, please state which authors and which funders.

1d) If you did not receive any funding for this study, please state: “The authors received no specific funding for this work.”

We have included the following paragraph at the end of the manuscript:

“The authors received no specific funding for this work. Data analysis conducted in this study was limited to secondary use of information previously collected in the course of normal care. No dedicated funding source was allocated for this study. The funder provided support in the form of salaries for all authors, but did not have any additional role in the study design, data collection and analysis, decision to publish, or preparation of the manuscript. The specific roles of these authors are articulated in the ‘author contributions’ section.” Pages 19-20, lines 272-277.

2. Thank you for providing the following Funding Statement:

I have read the journal's policy and the authors of this manuscript have the following competing interests: Competing interests

ML and GW are shareholders and directors of Oncologica UK Ltd. ML, KH, TH, RT and GW are currently employed at Oncologica UK Ltd.

We note that one or more of the authors is affiliated with the funding organization, indicating the funder may have had some role in the design, data collection, analysis or preparation of your manuscript for publication; in other words, the funder played an indirect role through the participation of the co-authors.

If the funding organization did not play a role in the study design, data collection and analysis, decision to publish, or preparation of the manuscript and only provided financial support in the form of authors' salaries and/or research materials, please review your statements relating to the author contributions, and ensure you have specifically and accurately indicated the role(s) that these authors had in your study in the Author Contributions section of the online submission form. Please make any necessary amendments directly within this section of the online submission form. Please also update your Funding Statement to include the following statement: “The funder provided support in the form of salaries for authors [insert relevant initials], but did not have any additional role in the study design, data collection and analysis, decision to publish, or preparation of the manuscript. The specific roles of these authors are articulated in the ‘author contributions’

section.”

If the funding organization did have an additional role, please state and explain that role within your Funding Statement.

Please also provide an updated Competing Interests Statement declaring this commercial affiliation along with any other relevant declarations relating to employment, consultancy, patents, products in development, or marketed products, etc.

We have clarified these points in the above paragraph which has been inserted at the end of the manuscript. Pages 19-20, lines 272-277.

3. We note that your Data Availability statement states the following: "The datasets analysed during the current study are provided in the Supplementary File. Further technical details are available from the corresponding author upon reasonable request."

PLOS journals require that all data presented in the study be made publicly available at or before the time of publication. If there are legal or ethical restrictions on the data being made publicly available, such as IRB restriction or patient confidentiality, authors must provide a way for fellow researchers to access the data.

At this time, please confirm that your submission contains your "minimal data set", which PLOS defines as consisting of the data set used to reach the conclusions drawn in the manuscript with related metadata and methods, and any additional data required to replicate the reported study findings in their entirety. This includes:

We can confirm that the submission contains the minimal dataset.

Values reported in Results for fusion frequencies, metastatic status and fusion read counts (page 7) derived from raw data in Supplementary Table 6, summarised for each cancer type in Supplementary Table 8. Age and gender data (page 8) is located in Supplementary Tables 2 and 9. Table 1 contains data used to report targeted therapy protocols (page 8).

Figures 1, 2 and 3 contain raw values in legend, with data derived from Supplementary tables 6 and 8.

3) The points extracted from images for analysis.

N/A

27 December 2021

Dear Dr Reddi, 

Many thanks for your email. We have now addressed your requests and wish to submit the revised manuscript. 

We look forward to your reply.

Best Wishes

Dr Marco Loddo

BSc PhD

Professor Gareth H Williams

BSc MBChB PhD FRCPath FLSW

PONE-D-21-02472R1

Utilisation of semiconductor sequencing for detection of actionable fusions in solid tumors Dr Marco Loddo

Dear Dr. Loddo,

We've checked your submission and before we can proceed, we need you to address the following issues:

1. Thank you for providing additional information. However, in order to provide our readers with as much transparency as possible, PLOS ONE policy requires that certain statements be present in the competing interests statement when authors have commercial affiliations.

At this time, we would like to propose the following updated funding disclosure statement to appear alongside your published paper:

"The authors received no specific funding for this work. Data analysis conducted in this study was limited to secondary use of information previously collected in the course of normal care. No dedicated funding source was allocated for this study. The funder provided support in the form of salaries for all authors, but did not have any additional role in the study design, data collection and analysis, decision to publish, or preparation of the manuscript. The specific roles of these authors are articulated in the ‘author contributions’ section.” Pages 19-20, lines 272-277."

I confirm this paragraph has already been inserted in the previous submission. 

We would also like to propose the following updated competing interest statement to appear alongside your published paper:

"The authors have read the journal's policy and the authors of this manuscript have the following competing interests: Authors ML, KH, AL, RT, and GW are currently salaried employees at Oncologica UK Ltd. TH was a salaried employee at the time of data generation for this manuscript. There are no patents, products in development, or marketed products associated with this research to declare. This does not alter our adherence to PLOS ONE policies on sharing data and materials."

The competing interest section has now been amended as requested

Please confirm or amend this statement to proceed. We hope to hear from you soon.

---

## [Decision Letter · Decision Letter 2]

25 Jul 2022

Utilisation of semiconductor sequencing for detection of actionable fusions in solid tumors

PONE-D-21-02472R2

Dear Dr. Loddo,

We’re pleased to inform you that your manuscript has been judged scientifically suitable for publication and will be formally accepted for publication once it meets all outstanding technical requirements.

Kind regards,

Honey V. Reddi

Academic Editor

PLOS ONE

Additional Editor Comments (optional):

We thank the authors for addressing the reviewer comments. The manuscript is now acceptable for publication.

Reviewers' comments:

Reviewer's Responses to Questions

**Comments to the Author**

1. If the authors have adequately addressed your comments raised in a previous round of review and you feel that this manuscript is now acceptable for publication, you may indicate that here to bypass the “Comments to the Author” section, enter your conflict of interest statement in the “Confidential to Editor” section, and submit your "Accept" recommendation.

Reviewer #1: All comments have been addressed

2. Is the manuscript technically sound, and do the data support the conclusions?

Reviewer #1: Yes

3. Has the statistical analysis been performed appropriately and rigorously? 

Reviewer #1: Yes

4. Have the authors made all data underlying the findings in their manuscript fully available?

Reviewer #1: Yes

5. Is the manuscript presented in an intelligible fashion and written in standard English?

Reviewer #1: Yes

6. Review Comments to the Author

Reviewer #1: Accept

7. PLOS authors have the option to publish the peer review history of their article (what does this mean?). If published, this will include your full peer review and any attached files.

Reviewer #1: No

---

## [Editor Report · Acceptance letter]

11 Aug 2022

PONE-D-21-02472R2 

Utilisation of semiconductor sequencing for detection of actionable fusions in solid tumours 

Dear Dr. Loddo:

I'm pleased to inform you that your manuscript has been deemed suitable for publication in PLOS ONE. Congratulations! Your manuscript is now with our production department. 

Kind regards, 

on behalf of

Dr. Honey V. Reddi 

Academic Editor

PLOS ONE